# Group B *Streptococcus* Early-Onset Disease: New Preventive and Diagnostic Tools to Decrease the Burden of Antibiotic Use

**DOI:** 10.3390/antibiotics12030489

**Published:** 2023-03-01

**Authors:** Charlotte M. Nusman, Linde Snoek, Lisanne M. van Leeuwen, Thomas H. Dierikx, Bo M. van der Weijden, Niek B. Achten, Merijn W. Bijlsma, Douwe H. Visser, Marlies A. van Houten, Vincent Bekker, Tim G. J. de Meij, Ellen van Rossem, Mariet Felderhof, Frans B. Plötz

**Affiliations:** 1Department of Paediatrics, Emma Children’s Hospital, Amsterdam University Medical Centre, Meibergdreef 9, 1105 AZ Amsterdam, The Netherlands; 2Department of Neurology, Amsterdam University Medical Centre, Meibergdreef 9, 1105 AZ Amsterdam, The Netherlands; 3Department of Paediatrics and Department of Vaccin, Infection and Immunology, Spaarne Hospital, Boerhaavelaan 22, 2035 RC Haarlem, The Netherlands; 4Department of Paediatrics, Willem Alexander Children Hospital, Leiden University Medical Center, Albinusdreef 2, 2333 ZA Leiden, The Netherlands; 5Department of Pediatric Gastroenterology, Emma Children’s Hospital, Amsterdam University Medical Centre, Meibergdreef 9, 1105 AZ Amsterdam, The Netherlands; 6Amsterdam Gastroenterology Endocrinology Metabolism Research Institute, Meibergdreef 69-71, 1105 BK Amsterdam, The Netherlands; 7Department of Paediatrics, Tergooi Hospital, Rijksstraatweg 1, 1261 AN Blaricum, The Netherlands; 8Department of Paediatrics, Erasmus University Medical Centre, Sophia Children’s Hospital, Wytemaweg 80, 3015 CN Rotterdam, The Netherlands; 9Department of Neonatology, Emma Children’s Hospital, Amsterdam University Medical Centre, Meibergdreef 9, 1105 AZ Amsterdam, The Netherlands; 10Division of Neonatology, Department of Pediatrics, Willem Alexander Children’s Hospital, Leiden University Medical Center, Albinusdreef 2, 2333 ZA Leiden, The Netherlands; 11Department of Paediatrics, Flevo Hospital, Hospitaalweg 1, 1315 RA Almere, The Netherlands

**Keywords:** antibiotics, biomarkers, blood culture, early-onset neonatal sepsis, Group B *Streptococcus*, guidelines, molecular culture techniques

## Abstract

The difficulty in recognizing early-onset neonatal sepsis (EONS) in a timely manner due to non-specific symptoms and the limitations of diagnostic tests, combined with the risk of serious consequences if EONS is not treated in a timely manner, has resulted in a low threshold for starting empirical antibiotic treatment. New guideline strategies, such as the neonatal sepsis calculator, have been proven to reduce the antibiotic burden related to EONS, but lack sensitivity for detecting EONS. In this review, the potential of novel, targeted preventive and diagnostic methods for EONS is discussed from three different perspectives: maternal, umbilical cord and newborn perspectives. Promising strategies from the maternal perspective include Group B *Streptococcus* (GBS) prevention, exploring the virulence factors of GBS, maternal immunization and antepartum biomarkers. The diagnostic methods obtained from the umbilical cord are preliminary but promising. Finally, promising fields from the newborn perspective include biomarkers, new microbiological techniques and clinical prediction and monitoring strategies. Consensus on the definition of EONS and the standardization of research on novel diagnostic biomarkers are crucial for future implementation and to reduce current antibiotic overexposure in newborns.

## 1. Introduction

Early-onset neonatal sepsis (EONS) is a rare but potentially life-threatening disease. It is defined as sepsis within 72 h after birth, confirmed with a positive blood and/or cerebrospinal fluid culture [1,2]. In Europe, the incidence of culture-proven EONS is estimated to be around 0.5–1 per 1000 live newborns [3]. The predominant pathogen is Group B *Streptococcus* (GBS; *Streptococcus agalactiae*), which causes one-third to half of all EONS cases, followed by *Escherichia coli* (*E. coli*) [4,5].

The difficulty in recognizing EONS in a timely manner due to non-specific symptoms and limitations in the time-effectiveness and accuracy of diagnostic testing, combined with the risk of serious consequences if EONS is not treated in a timely manner, has resulted in a low threshold for starting empirical antibiotic treatment. This has led to significant overtreatment, despite the well-known short- and long-term side effects [6,7,8,9,10,11,12]. In addition, despite a negative blood culture, antibiotic therapy is continued for more than three days in about 30% of newborns once started [13,14]. Suspicion of the quality of the blood culture, for example based on incorrectly obtained cultures, low sensitivity and the use of intrapartum antibiotic prophylaxis, are reported reasons for continuing antibiotic therapy [15]. Consequently, culture-negative sepsis is frequently diagnosed, although the actual incidence or even the existence of such a phenomenon is uncertain, as plenty of studies have used prolonged antibiotic treatment as a diagnostic criterion.

The current guidelines contain three general approaches to identify newborns with an increased risk of EONS: a categorical risk factor assessment, a multivariate risk assessment (early-onset sepsis calculator) and a risk assessment primarily based on newborn clinical conditions [16]. A meta-analysis showed that the implementation of the neonatal sepsis calculator reduced the use of empirical antibiotic treatment, with a relative risk of 56% when compared to categorical approaches [17]. However, none of the current approaches have the capacity to predict EONS with acceptably high sensitivity [18,19]. Therefore, improved EONS risk stratification methods are needed. In addition, the categorical approach is the only guideline that provides recommendations for intrapartum antibiotic prophylaxis in order to prevent GBS-EONS. Despite these recommendations, the incidence of GBS-EONS is increasing [20,21].

In this article, we discuss our research expertise and review the available evidence to develop novel strategies aimed at preventing GBS-EONS and increasing diagnostic sensitivity for detecting EONS. We approach this from maternal, umbilical cord and neonatal perspectives, including their potential to decrease antibiotic therapy and to be incorporated into future guidelines (Figure 1).

## 2. Maternal Perspective

### 2.1. GBS Prevention

Most countries use a screening strategy that recommends screening pregnant women between 35 and 37 weeks of gestation for GBS carriage [22]. Those who are GBS carriers receive intrapartum antibiotic prophylaxis (IAP) [23]. Countries that have introduced such a screening strategy have observed a substantial reduction in EONS. In the case of risk-based screening, only pregnant women with certain risk factors are eligible for either screening or receiving IAP directly [24,25]. However, unexpectedly, these countries have not observed a decline in early-onset GBS disease [20,21]. The limited sensitivity of risk-based guidelines may explain their lack of effectiveness. Firstly, up to one-fifth of pregnant GBS carriers have no risk factors that would lead to IAP recommendations [26]. Secondly, culture results may be falsely negative, especially when the culture guidelines for the detection and identification of GBS are not followed [27,28]. These guidelines recommend sampling both the lower vagina and the rectum using a flocked swab and using adequate transport and enriched culture media in order to optimize sensitivity [27]. Thirdly, it usually takes 24 to 48 h for culture results to become available, which can lead to inadequate prophylaxis. Fourthly, there is low compliance to GBS risk-based guidelines [29]. False-negative culture results can also occur using the screening strategy. GBS carriage can be intermittent. For example, between 5% and 7% of women who test negative for GBS at 35–37 weeks of gestation are positive at the time of delivery [28,30]. Therefore, it is important to minimize the time between sampling and delivery by adhering to guidelines that recommend screening between 35–37 weeks of gestation [28].

The sensitivity of IAP guidelines could be increased using a screening-based approach, instead of a risk-based approach, in combination with bedside polymerase chain reaction (PCR) tests that can detect GBS colonization during labor [31]. These tests provide results within hours after sampling instead of days, thereby facilitating the timely administration of IAP [26,32]. Previous studies have demonstrated similar or higher sensitivity of PCR tests for the detection of GBS colonization compared to screening- and risk factor-based culture mehtods [26,32]. Obviously, increased adherence to IAP guidelines by optimizing implementation would also result in higher sensitivity.

### 2.2. Host–Pathogen Interactions in GBS Disease

The specificity of both screening- and risk-based IAP guidelines is limited, leading to high rates of unnecessary antibiotic treatment in women. Whether GBS transmission leads to harmless colonization or to invasive disease depends on the invasive potential of the GBS bacterium and the susceptibility of the hosts, i.e., the mother and her child. GBS express virulence factors to invade tissue barriers, evade host defense mechanisms and cause damage to host tissues in order to cause disease. Examples of virulence factors are α-like proteins that enable cell invasion, the polysaccharide capsule that causes host immune evasion and the hyper-virulent GBS surface-anchored adhesin protein (HvgA) that enables crossing of the blood–brain barrier [33,34]. These virulence factors are coded in bacterial virulence genes, and these genes vary among GBS subtypes. Epidemiological research has consistently shown that specific GBS bacterial genotypes, such as CC17, are over-represented in invasive disease compared with colonization, strongly suggesting that invasive GBS strains have unique bacterial virulence genes [20,35,36,37,38,39,40]. The specificity of IAP strategies could possibly be improved not only by screening for GBS carriage, but also by incorporating multiple genetic markers into a single PCR test, making it possible to determine both the presence and the invasive potential of GBS bacteria [37]. More extensive studies are needed in order to determine the diagnostic performance of these genetic virulence markers and to evaluate their potential in combination with current national guidelines for EONS prevention or the neonatal sepsis calculator.

On the host side, it is well established that newborns with low levels of protective antibodies against GBS virulence factors are more susceptible to developing invasive GBS disease [22]. In the final months of pregnancy, antibodies are actively transported across the placenta from the mother to the fetus. Newborns whose mothers have low levels of GBS-protective antibodies or newborns born preterm are at higher risk of invasive GBS disease [41,42,43]. Screening for antibodies against infectious diseases is routinely performed during pregnancy. Measuring maternal IgG antibody concentrations against common GBS virulence factors in blood from the mother and cord blood from the baby could allow for the more precise identification of newborns that are susceptible to developing invasive GBS disease and might improve targeted GBS prophylaxis. Hypothetically, one could measure antibody levels against GBS virulence factors during pregnancy, followed by a bedside PCR test in order to determine the presence of GBS and GBS subtypes. Carriers of a hyper-virulent GBS strain with low antibody levels against that strain could receive IAP, whereas women with sufficient antibody levels and a less invasive type of strain would not need IAP. Bacterial genome-wide association studies used to identify new genetic GBS virulence markers are becoming available, and IgG antibody concentrations that protect against invasive GBS disease are being determined [22,38,44]. Future studies should determine the diagnostic value of these genetic virulence markers and serological antibodies in order to identify the mothers of newborns who are at high and low risk of EONS.

### 2.3. Maternal Immunization

Another promising strategy to prevent invasive GBS disease is maternal immunization during pregnancy. Lower antibody titers against the GBS polysaccharide capsule and some GBS surface proteins in uncolonized pregnant women are associated with a higher probability of becoming colonized during pregnancy [45,46]. Vaccination may prevent the transmission of GBS from mother to child by reducing GBS carriage [47]. More importantly, the transfer of protective IgG antibodies to the newborn via the placenta might protect the child from invasive GBS disease during the first months of life. A hexavalent GBS conjugate vaccine was recently proven safe and immunogenic in healthy adults and is currently being studied in pregnant women [48]. However, clinical efficacy studies are time-consuming and expensive due to low disease incidence. Since antibody levels are related to the risk of invasive disease, determining the serological immune correlates of protection is a promising approach that may accelerate the licensure of a maternal GBS vaccine [49]. However, maternal GBS immunization does not prevent GBS disease in very preterm children since the active transfer of IgG only starts in the third trimester. Therefore, adequate EONS risk stratification methods remain necessary.

### 2.4. Antepartum Immunological Biomarkers

Chorioamnionitis is a heterogeneous condition characterized by intrauterine infection, inflammation or both and is a common cause of preterm birth and adverse neonatal outcomes. Nevertheless, the actual intrauterine transmission of micro-organisms to the newborn that result in EONS is limited, and the risk of EONS in chorioamnionitis cases in well-appearing newborns after birth is <1% [50,51]. In an attempt to differentiate between infectious and non-infectious causes of maternal fever and causes of amniotic fluid infection that could result in EONS, experts introduced the term ‘triple I’ to aid clinical decision making. Triple I (infection, inflammation or both) is based on clinical characteristics, maternal white blood cell counts, bacterial cultures and the histological evidence of placental infection and/or fetal membranes [52]. Maternal immunological biomarkers that differentiate between inflammation and infection are potentially interesting diagnostic targets to identify those at risk. In addition, the material is relatively easy to obtain, and by investigating the mother’s condition, there is the potential to start appropriate treatment early in the course of the disease. These biomarkers are of particular interest in women with preterm labor who are being evaluated for tocolysis.

The different biomarkers evaluated so far lack diagnostic accuracy, and therefore could only be used as part of a screening method instead of as a diagnostic tool. The immunological biomarkers studied include, among others, hematological cell indices, C-reactive protein (CRP), Tumor necrosis factor α (TNF-α) and interleukins. CRP in maternal serum is one of the most frequently studied markers used to identify newborns who are at risk of EONS. However, large variations in diagnostic accuracy have been observed, with sensitivity ranging from 21% to 94% and specificity ranging from 48% to 95% [53]. This is explained by the large variations in study design and the increase in CRP that also occurs in response to many non-infectious conditions [54]. A non-invasive alternative that has been explored by several researchers is IL-6 testing in cervicovaginal or amniotic fluid. This qualitative test provides the clinician with a quick and easy-to-interpret answer about the presence of intrauterine inflammatory processes. However, the diagnostic performance regarding EONS detection is poor, with widely ranging diagnostic accuracy, depending on test specifics and cut-off values [53].

## 3. Umbilical Cord Perspective

### 3.1. Biomarkers

The use of umbilical cord blood biomarkers to diagnose newborns with EONS could provide information early in the course of the disease, even before clinical signs are apparent. Postnatal umbilical cord sampling is non-invasive, painless and easy to perform, and a larger volume of blood is available for testing than in the case of sampling postpartum in the newborn. This is particularly important for neonates born with an extremely low birth weight. A large range of biomarkers have been tested in the umbilical cord blood of preterm and term newborns, for example, CRP, procalcitonin (PCT), interleukins, TNF-α, interferon gamma (IFN-y), serum amyloid A (SAA), presepsin, etc. [48,49,50,51,52,53]. Similar to maternal serum samples, CRP does not perform well in distinguishing between infected and non-infected newborns, with low sensitivity (around 50%) across studies [53]. The specificity and negative predictive value of CRP are slightly better, meaning that low CRP levels might be helpful in identifying newborns who are at low risk of EONS. More interesting biomarkers include PCT and IL-6, which have performed better than CRP and most of the tested interleukins in the majority of studies [55,56]. Recent meta-analyses have shown a pooled sensitivity >80% and a pooled specificity >92% for PCT (in eight studies), and a pooled sensitivity of 76–78% and a pooled specificity of 79–82% for IL-6 [53,57,58]. The combination of PCT and IL-6 is less consistent but still promising, with sensitivity ranging between 46% and 91% and specificity between 77% and 99%. Moreover, subgroup analyses show that IL-6 performs even better in premature newborns than in term newborns, with sensitivity and specificity >95% [55].

Promising biomarkers that have been studied in small groups using blood from the umbilical cord include presepsin in term and preterm newborns, SAA and haptoglobin in preterm newborns and extracellular heat-shock proteins (eHsps) 60 and 70. Larger clinical studies are necessary in order to confirm their diagnostic use in term and preterm newborns [59,60,61].

### 3.2. Blood Culture and Molecular Techniques

Although conventional peripheral bacterial cultures are considered the gold standard in order to establish a diagnosis of EONS, its sensitivity has been questioned. The majority of infants treated with antibiotics for presumed EONS have negative blood cultures and have so-called culture-negative EONS [62]. Multiple factors may influence the risk of false-negative blood cultures, and thus impact sensitivity. Low bacterial loads in infants with EONS, inadequate blood volume collection and exposure to maternal intrapartum antibiotics may increase the risk of false-negative blood cultures, which can decrease sensitivity and challenge the reliability of peripheral bacterial culture as the gold standard [63]. The American Academy of Pediatrics guidelines for the evaluation of early- and late-onset sepsis recommend obtaining a minimum blood sample of 1 mL, as there is a direct correlation between the volume of blood obtained and the sensitivity of the blood culture [64]. In line with these recommendations is a recent review by Huber et al. on the current approaches for collecting adequate blood volumes for paediatric blood cultures [65]. They concluded that a sample of 1.0 to 1.5 mL for children weighing less than 11 kg seems most appropriate [66]. Furthermore, increasing the volume of blood does not increase sensitivity. Ohnishi et al. showed that the positive detection rate of blood cultures did not improve even when the collected blood volume was increased from a median blood volume of 1.64 mL to 2.41 mL [67]. Following this observation, the authors concluded that one millilitre of blood may be adequate for infants and children [65,66,67].

Standard operating procedures have been developed to collect umbilical cord blood in a sterile manner in order to minimize the risk of contamination during sampling [68]. A meta-analysis demonstrated that cultures using cord blood were negative in 91% of infants with negative peripheral blood cultures and positive in 75% of infants with positive peripheral blood cultures [69]. As peripheral blood cultures are incorporated in the consensus definition of EONS (culture-proven EONS), it is difficult to determine which culture is a false positive/negative when there are discrepancies in the results between cord blood cultures and peripheral blood cultures. In the same review, a limited number of studies compared the diagnostic accuracy of both tests for clinical sepsis. Peripheral blood cultures had a sensitivity of 20.5% and a specificity of 100%, compared with a sensitivity and specificity of 43% and 98%, respectively, for cord blood cultures for clinical EONS. However, these studies included a limited number of infants and were at a high risk of bias. Thus, future larger studies are warranted in order to determine the usability of cord blood cultures in EONS.

## 4. Newborn Perspective

### 4.1. Hematological Cell Indices

Hematological cell indices have been used for decades in order to aid the diagnosis of EONS, but their diagnostic performance as individual markers is poor. In general, indices, such as complete blood cell counts, show low sensitivity and relatively high specificity, meaning that sepsis could not be ruled out in cases of normal cell counts [53,70]. Platelets have recently gained renewed interest as first-line responders to infection and as biomarkers for EONS. Thrombocytopenia is seen in almost half of the NICU patients with early- or late-onset sepsis. Moreover, an increased mean platelet volume (MPV) has been observed in patients with EONS compared with healthy controls [71,72].

### 4.2. Biomarkers

The perfect biomarker should be able to differentiate between newborns with an infection and those with symptoms associated with the transition after birth that mimics EONS. Additionally, a biomarker should identify a well-appearing newborn without maternal risk factors that will develop EONS hours after birth, and the results should be available as early as possible. Testing for host markers that change in response to infection would require a smaller blood volume and would result in quicker results than culture-based methods. Various systematic reviews show that a single biomarker with optimal diagnostic accuracy does not exist to date, but numerous researchers have attempted to find one [73,74]. The enormous number of studies focusing on a single biomarker to aid in the diagnosis of EONS has led to the realization that one biomarker is not sufficient. Even though numerous promising markers exist, their evaluation and validation in large cohorts are often missing. Meta-analyses are hampered by the large heterogeneity in case definitions, cut-off values and the techniques used. Novel approaches include biomarker combinations or -omics techniques that identify host response profiles. With the rapid development of multiplexed platforms and point-of-care systems, the determination of multiple biomarkers at once is expected to become more accessible [75,76].

Among the most commonly used and studied biomarkers are acute-phase proteins CRP, PCT and IL-6. CRP performs poorly as an individual marker in diagnosing EONS early in the course of the disease. PCT is described to be less reliable after antepartum antibiotics are administered, and widely ranging sensitivity and specificity have been described [53,77,78]. IL-6 shows diagnostic accuracy, with a reported pooled sensitivity of 72% and a pooled specificity of 75%, which makes it not satisfactory enough to use as a single diagnostic marker [53].

A new and promising biomarker is presepsin, which is a soluble CD14 subtype that is released by monocytes and macrophages in response to bacterial infection. It is not affected by perinatal factors or gestational age. Three recent meta-analyses reported that presepsin has a pooled sensitivity ranging between 81% and 93% and a pooled specificity between 86% and 91% [79,80,81]. Another good marker is SAA, which is an acute-phase protein produced in response to IL-6. Its diagnostic accuracy is reported to be high at the onset of symptoms, with a sensitivity and specificity >90%, but large studies solely focusing on newborns who are at risk of EONS are lacking [53,70].

### 4.3. Blood Culture and Molecular Techniques

The two major drawbacks of conventional peripheral blood cultures include a delay in the availability of results of up to 48–72 h following sampling and the questioned sensitivity, as discussed above. Inertia prevents the traditional blood culture from serving as a diagnostic test to exclude EONS directly postpartum [82]. Rapid diagnostic tools with high sensitivity and a negative predictive value within hours after birth could result in a significant reduction in the overuse of antibiotics in uninfected newborns. Molecular techniques, such as broad-range PCR, real-time PCR and multiplex PCR, generate results faster than conventional blood cultures (Figure 2). Additionally, these molecular techniques are not influenced by maternal IAP, may detect species that are undetectable using conventional cultures and might detect pathogens with low loads in a sample; thus, they are expected to have higher sensitivity [82]. A Cochrane review demonstrated a sensitivity of 94% and a specificity of 92% for these molecular assays in studies including cases of both culture-proven EONS and late-onset neonatal sepsis. Insufficient numbers of EONS cases were included in these studies to perform a subgroup analysis for this specific entity [83]. Again, regarding discrepancies in the results between molecular cultures and conventional blood cultures, it may be challenging to determine which of the tests is a false negative/positive, particularly since traditional blood cultures are still considered the gold standard. This is underlined by the results of two recent studies that demonstrated more positive molecular cultures than positive conventional cultures in both clinical EONS cases and controls [84,85].

However, these PCR techniques are restricted by the panel used and can only detect a predefined set of pathogens. Unrestricted techniques, such as 16S sequencing, on the other hand, are costly and have a reporting delay of one to several days, which is comparable to conventional culturing. A novel technique called Molecular Culture using IS-pro (inBiome, Amsterdam, the Netherlands) is an unrestricted PCR-based technique that allows the profiling of all the bacterial species present in a sample to be conducted within 5 h, but its potential has not yet been investigated in EONS [86]. In a small cohort study performed by our group that included 40 samples from newborns suspected to have EONS, conventional blood cultures were negative for all the participants, including 17 clinical EONS cases. The Molecular Culture results were identical in 92.5% of the samples and the identified micro-organisms were missed by conventional culture in three samples. As this was a small cohort and did not include culture-positive EONS cases, future larger studies are needed in order to determine whether molecular profiling can predict conventional culture results, and thus may guide the clinicians’ antibiotic stewardship faster than conventional culture methods (unpublished data, personal communication).

If one aims to confirm that other culture techniques, such as umbilical cord blood cultures or molecular culture techniques, have higher sensitivity to EONS than conventional peripheral cultures, a predefined consensus definition of clinical EONS is pivotal [87]. Future larger studies could then compare the diagnostic accuracy of conventional peripheral blood culture methods and of these other detection techniques for EONS using this predefined definition.

### 4.4. Clinical Prediction and Monitoring

Major advancements towards the better use of antibiotics postpartum have been realized with the development of the neonatal sepsis calculator by the research division of Kaiser Permanente in the United States [88,89]. Based on a large dataset, objective clinical maternal and neonatal parameters are used in an accessible clinical risk calculator that provides guidance on the use of antibiotics after birth. This method is associated with a safe and effective reduction in the rate of antibiotic treatment after birth [17]. However, post-development analyses and data evaluating this model suggest an untapped potential for the improvement of this type of clinical monitoring and related risk stratification; the overtreatment of non-infected newborns with antibiotics remains significant, whereas the sepsis calculator does not identify EONS cases better than the alternative methods [18,19,90]. Fortunately, there are established methods to evaluate and recalibrate clinical prediction tools in order to improve the accuracy of risk estimations, and thus the allocation of antibiotics [91,92]. We have recently developed, validated and published an open-source version of the EOS calculator, which enables electronic integration and may allow for the large-scale and setting-specific calibration of this tool. In addition, in low- and middle-income countries (LMICs), mobile phones are increasingly accessible, and since the EOS calculator is an online tool, the calculator may also be applicable in LMICs. However, validation of the EOS calculator in this specific population is a requirement.

Data obtained from the development and evaluation of the sepsis calculator demonstrate the importance of neonatal clinical symptoms for both ruling out and early detection of EONS [90,93]. Therefore, the careful observation of neonatal clinical parameters is crucial for the correct use of the neonatal EOS calculator. In line with these data, some centers are now opting for exclusive ‘serial physical examinations’ as a strategy to allocate antibiotics at birth [94,95]. In these strategies, newborns receive structural and protocolized physical examinations, and only persistently symptomatic newborns receive antibiotics. Evidence regarding this practice is scarce but promising in terms of reducing antibiotic exposure rates and unnecessary laboratory evaluations. Moreover, it appears that this practice does not worsen the outcome of neonates at risk or those presenting with mild, equivocal or transient symptoms [96]. The strategy is condoned by the American Academy of Pediatrics for newborns born at ≥35 weeks of gestational age [97]. However, the current lack of a standardized approach regarding the interval, content and healthcare provider responsible for the examination, as well as the threshold definition and resources needed for this strategy, may be problematic. The role of maternal risk factors, which may have a modest but clear impact on EONS risk, is yet to be defined in this approach. Moreover, we should not forget that the difficulty in clinically distinguishing EONS from other causes of symptomatology has led to the widespread overtreatment of newborns that we see today.

Therefore, clinical risk prediction and monitoring may currently be the most effective tools available to reduce overtreatment with antibiotics in newborns, but they still fall short in terms of diagnostic accuracy. New iterations of the EOS calculator tool and serial physical examinations will likely lead to some improvement, but they will undoubtedly require the simultaneous use of biomarkers and improved diagnostic microbiology tools in order to reach the goal of treating every newborn with EONS in a timely manner, without overtreating those without EONS.

## 5. Conclusions

In order to further decrease the number of newborns who are empirically treated with antibiotic therapy and to reduce the duration of antibiotic therapy in the case of a negative blood culture, it is necessary to develop novel strategies aimed at preventing EONS and at increasing diagnostic sensitivity for detecting EONS. Promising strategies from the maternal perspective include GBS prevention, exploring the virulence factors of GBS, maternal immunization and antepartum biomarkers. The diagnostic methods obtained from the umbilical cord are preliminary and, although promising, in general not yet ready for implementation in daily practice. The most promising and extensively studied fields from the newborn perspective are biomarkers, new microbiological techniques and clinical prediction and monitoring strategies. Finally, future research should also focus on the possibility of using combinations of markers in samples from different sources. As mentioned throughout this entire review, consensus on the definition of EONS and the standardization of research on novel biomarkers are crucial for the further implementation of new diagnostic strategies and for reducing the antibiotic burden in newborns [87].

## Figures and Tables

**Figure 1 antibiotics-12-00489-f001:**
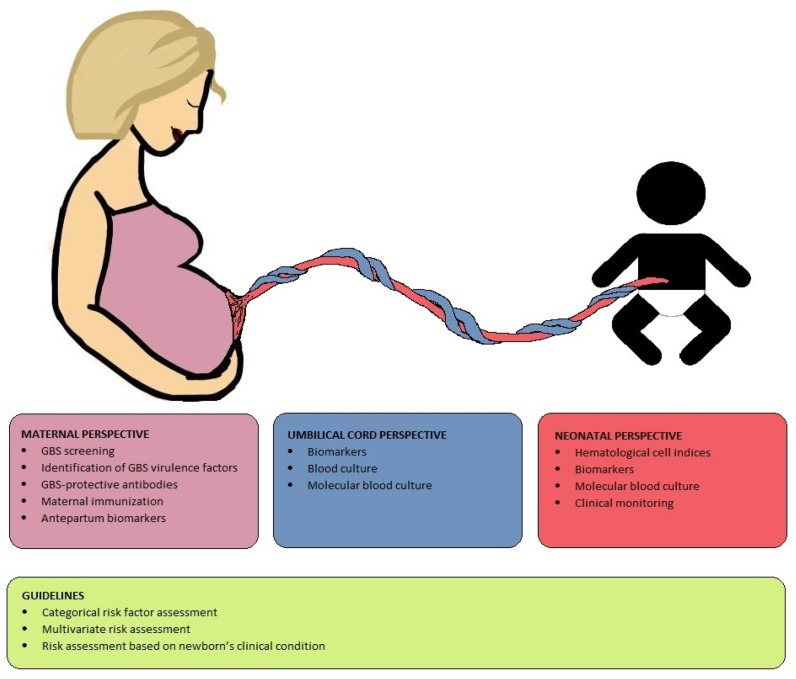
Overview of diagnostic and preventive strategies for EONS from three different perspectives, complemented with the various assessment methods ofthe guidelines.

**Figure 2 antibiotics-12-00489-f002:**
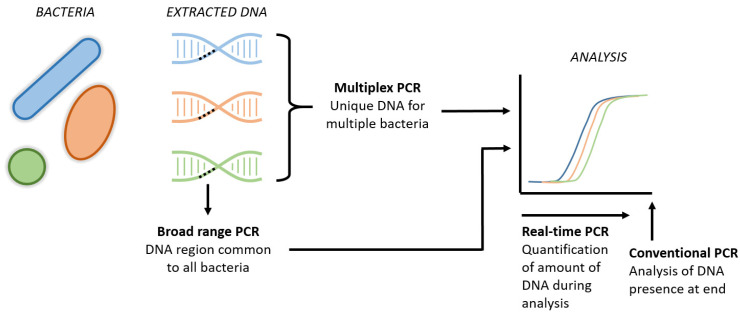
Schematic visualization of new PCR techniques. The broad range PCRs refers to the analysis of a common region present in all bacteria, aiming to detect the presence of bacterial DNA in general. Multiplex PCR includes the simultaneous DNA analysis of several specific bacteria in one tube. Real-time PCR is the new technique that enables us to quantify the amount of DNA throughout the entire analysis, instead of the presence of DNA at the end of the analysis in conventional PCR.

## Data Availability

No new data were created or analyzed in this study. Data sharing is not applicable to this article.

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
