# Peer review of "Group B Streptococcus Early-Onset Disease: New Preventive and Diagnostic Tools to Decrease the Burden of Antibiotic Use"

_antibiotics, 2023, doi:10.3390/antibiotics12030489_

Round 1
Reviewer 1 Report
Dear authors,
Thanks for your efforts to contribute to the body of literature on EONS- an topic surrounded by many controversies in clinical practice. In the text, you described different aspects from different perspectives (maternal, umbilical cord, neonatal vs. guidelines). I appreciate this effort in structuring information. Yet, in the present state, this manuscript is not a valuable addition to the existing literature. Therefore, I recommend a thorough re-conceptualization and revision of this manuscript. I will give you some recommendations in the forthcoming section.
1) Previous work on this topic
Multiple high-quality reviews have been published recently, e.g.:
-
Early onset neonatal sepsis: diagnostic dilemmas and practical management.- Glaser MA, Hughes LM, Jnah A, Newberry D. Neonatal Sepsis: A Review of Pathophysiology and Current Management Strategies. Adv Neonatal Care. 2021 Feb 1;21(1):49-60. doi: 10.1097/ANC.0000000000000769. PMID: 32956076.
- Simonsen KA, Anderson-Berry AL, Delair SF, Davies HD. Early-onset neonatal sepsis. Clin Microbiol Rev. 2014 Jan;27(1):21-47. doi: 10.1128/CMR.00031-13. PMID: 24396135; PMCID: PMC3910904.
- , et al. What’s new in the management of neonatal early-onset sepsis?
Therefore, I recommend you to start by reading these and identify unique topics that you could address in your manuscript.
2) Introduction
The introduction is too long and the first sentence is exactly the same as the abstract. That should be avoided.
3) Guidelines
There are many guidelines worldwide on this topic. Though, you elected to include two guidelines that are very similar- from a specific part of the world. You may therefore consider to submit this paper at a more local journal. If not, it is of utmost importance to include other guidelines and to resonate this in the remainder of the text (e.g., CPS, AAP, South Australia. I am sure that there will be guidelines/position statements in non-English speaking countries too).
4) Maternal perspective
Though EONS is in the title, I only read about GBS in this section. I appreciate this is the most common pathogen, but other bacteria responsible for EONS should not be ignored. Also, this section is disproportionally long and should therefore be condensed.
5) Newborn
There are separate (systematic) review papers on the diagnostic value on biomarkers for the (early) detection of EONS; as you allude to in lines 345-353. Hence, this section is too long. Instead, I would recommend to focus on clinical features and early warning scores. Furthermore, do you have any recommendations for colleagues practising in LMIC, where biomarkers and diagnostic alghorhitms may not be available?
6) Readability
I would respectfully like to mention that, although I think all authors have a decent command of English, the text is not very readable. Many parts of the texts need condensing. An English proofread service may be helpful. Furthermore, as per the instructions for authors, there should be at least two tables/graphs/figures. These are missing in the current state.
I greatly appreciate the efforts of this large group of authors with expertise in different subspecialties (gen peds, neurology, ID, gastro, neonatal). Yet, I think adding authors with expertise in reproductive ID/gyne, microbiology and pediatric infectious diseases would be beneficial.
Author Response
Reviewer #1
Dear authors,
Thanks for your efforts to contribute to the body of literature on EONS- an topic surrounded by many controversies in clinical practice. In the text, you described different aspects from different perspectives (maternal, umbilical cord, neonatal vs. guidelines). I appreciate this effort in structuring information. Yet, in the present state, this manuscript is not a valuable addition to the existing literature. Therefore, I recommend a thorough re-conceptualization and revision of this manuscript. I will give you some recommendations in the forthcoming section.
REPLY: Dear reviewer, thank you for the time and careful attention given to the initial submission by you and suggested recommendation how to improve our manuscript. We have responded to each of your comments separately. We do like to make a general comment. Our manuscript was reviewed by two reviewers (including you) and the overall comments varied significantly: “recommend a thorough re-conceptualization and revision of this manuscript” (yours) versus “enjoyed reading this very well written article” (second reviewer). Therefore, it was a challenge to address all the comments in the revised manuscript. We hope that you understand this dilemma.
We decided to use the same format, hence discuss strategies from a maternal, umbilical cord, and neonatal perspective. The reason for this is that all co-authors have their specific expertise and research knowledge. They all wrote the specific parts of the manuscript and explains why we focus on these parts of neonatal EONS management and the resulting broad narrative review. We also added this at the end of the introduction in the revised manuscript. Where possible, we refer to the review articles recommended by the first reviewer, and consequently parts of that specific section were condensed.
NEW: “In this article we will discuss our research expertise and available evidence regarding the currently applied and the accuracy of novel targeted diagnostic and preventive strategies for EONS.”
1) Previous work on this topic
Multiple high-quality reviews have been published recently, e.g.:
- Bedford Russell AR, Kumar R. Early onset neonatal sepsis: diagnostic dilemmas and practical management. Archives of Disease in Childhood - Fetal and Neonatal Edition 2015;100:F350-F354.
- Glaser MA, Hughes LM, Jnah A, Newberry D. Neonatal Sepsis: A Review of Pathophysiology and Current Management Strategies. Adv Neonatal Care. 2021 Feb 1;21(1):49-60. doi: 10.1097/ANC.0000000000000769. PMID: 32956076.
- Simonsen KA, Anderson-Berry AL, Delair SF, Davies HD. Early-onset neonatal sepsis. Clin Microbiol Rev. 2014 Jan;27(1):21-47. doi: 10.1128/CMR.00031-13. PMID: 24396135; PMCID: PMC3910904.
- Fleiss N, Schwabenbauer K, Randis TM, et al. What’s new in the management of neonatal early-onset sepsis? Archives of Disease in Childhood - Fetal and Neonatal Edition 2023;108:10-14.
Therefore, I recommend you to start by reading these and identify unique topics that you could address in your manuscript.
REPLY: We thank the reviewer for this suggestions and the provided review references. We were familiar with these review articles. We incorporated these references to shorten some sections were there was overlap in our manuscript.
NEW:
-Introduction:
Added reference: Simonsen KA, Anderson-Berry AL, Delair SF, Davies HD. Early-onset neonatal sepsis. Clin Microbiol Rev. 2014 Jan;27(1):21-47. doi: 10.1128/CMR.00031-13. PMID: 24396135; PMCID: PMC3910904.
- Guidelines paragraph: Please see also rebuttal comment #3.
Fleiss N, Schwabenbauer K, Randis TM, et al. What’s new in the management of neonatal early-onset sepsis? Archives of Disease in Childhood - Fetal and Neonatal Edition 2023;108:10-14.
-GBS prevention: routine vs risk based screening. Please see also rebuttal comment #4.
Bedford Russell AR, Kumar R. Early onset neonatal sepsis: diagnostic dilemmas and practical management. Archives of Disease in Childhood - Fetal and Neonatal Edition 2015;100:F350-F354.
2) Introduction
The introduction is too long and the first sentence is exactly the same as the abstract. That should be avoided.
REPLY: According to the reviewers suggestion we have deleted the first sentence in the abstract. We shortened the introduction by almost 30%. In particular, we skipped the part on the consequences of antibiotic use.
3) Guidelines
There are many guidelines worldwide on this topic. Though, you elected to include two guidelines that are very similar- from a specific part of the world. You may therefore consider to submit this paper at a more local journal. If not, it is of utmost importance to include other guidelines and to resonate this in the remainder of the text (e.g., CPS, AAP, South Australia. I am sure that there will be guidelines/position statements in non-English speaking countries too).
REPLY: We agree that there are many guidelines worldwide. However, it is not our intention to discuss and compare all these guidelines. In this section we emphasize that 3 general approaches are applied to identify newborns at increased risk for EONS but that none of these approaches have the capacity to predict EONS with an acceptable high sensitivity. These approaches are part of guidelines regarding EONS management. We significantly condensed this section and also refer to the review of Fleiss et al. We feel it is now more appropriate to incorporate this section in the introduction.
NEW: “Current guidelines contain three general approaches to identify newborns at increased risk of EONS: a categorical risk factor assessment, a multivariate risk assessment (the early-onset sepsis calculator), and risk assessment primarily based on the newborns clinical condition [Fleiss]. A meta-analysis showed that implementation of the neonatal sepsis calculator reduced the use of empirical antibiotic treatment with a relative risk of 56% compared to categorical approaches [21]. However, none of the current approaches have the capacity to predict EONS with an acceptable high sensitivity [22,23]. Therefore, improved EONS risk stratification methods are needed. “
4) Maternal perspective
Though EONS is in the title, I only read about GBS in this section. I appreciate this is the most common pathogen, but other bacteria responsible for EONS should not be ignored. Also, this section is disproportionally long and should therefore be condensed.
REPLY: The reviewer is correct regarding the title. We changed the title.
NEW: “Early-onset neonatal sepsis: new preventive and diagnostic tools to decrease the burden of antibiotic use”
REPLY: We agree with the reviewer that other bacteria are also responsible for EONS. However, the scope of our review is on novel preventive and diagnostics strategies, based on our research experiences combined with best available evidence. Our focus is therefore on GBS.
We condensed the section on GBS screening and refer to the review article of Bedford Russell AR. The sections on host-pathogen, maternal immunization, and antepartum immunological biomarkers were not addressed in the recommended review articles and remained therefore largely unchanged.
5) Newborn
There are separate (systematic) review papers on the diagnostic value on biomarkers for the (early) detection of EONS; as you allude to in lines 345-353. Hence, this section is too long.
REPLY: We agree with the reviewer that a number of nice review papers on biomarkers during EONS have been published. We included two of them and refer to these reviews. Further, we shortened this section.
Instead, I would recommend to focus on clinical features and early warning scores.
REPLY: In the section “Clinical prediction and monitoring” we discuss in depht that careful observation of neonatal clinical parameters is at the cornerstone of correct use of the neonatal EOS calculator. In line with these data, some centers are now opting for exclusive ‘serial physical examinations’ as a strategy to allocate antibiotics at birth. However, we also mention the drawbacks of solely clinical observation as advocated by serial physical examination approach. We added an extra reference (Berrardi 2016).
Furthermore, do you have any recommendations for colleagues practicing in LMIC, where biomarkers and diagnostic algorithms may not be available?
REPLY: This is an important remark. Because laboratory tests require technical expertise, equipment, electricity and refrigeration, they are often inaccessible in LMIC, which poses a serious problem for these proposed preventive and diagnostic tools mentioned in our manuscript. Hence, we believe that close clinical monitoring of the newborns remains the cornerstone in LMIC. In addition, the EOS calculator tool is online, and possible in the near future also as an app available on mobile phones, and could be an alternative. In contrast to lack of technical assistance, mobile phones are increasingly accessible. However, validation of the EOS calculator for this specific population is a requirement.
NEW: In addition, in low and middle income countries (LMIC) mobile phones are increasingly accessible and since the EOS calculator is an online tool, the calculator may also applicable in LMIC. However, validation of the EOS calculator for this specific population is a requirement.
6) Readability
I would respectfully like to mention that, although I think all authors have a decent command of English, the text is not very readable. Many parts of the texts need condensing. An English proofread service may be helpful. Furthermore, as per the instructions for authors, there should be at least two tables/graphs/figures. These are missing in the current state.
REPLY: We followed this suggestion and the revised version of this manuscript was sent forward to editing services of the Journal. Furthermore, we added a 2 figures.
I greatly appreciate the efforts of this large group of authors with expertise in different subspecialties (gen peds, neurology, ID, gastro, neonatal). Yet, I think adding authors with expertise in reproductive ID/gyne, microbiology and pediatric infectious diseases would be beneficial.
REPLY: We thank the reviewer for this suggestion. Please see also our general comment. The listed co-authors all have their specific expertise and research knowledge and are part of our neonatal EOS working group. For this invited review they all wrote the specific parts of the manuscript. That is why we focus on these parts of neonatal EONS management. We also added this at the end of the introduction in the revised manuscript.
NEW: “In this article we will discuss our research expertise and available evidence regarding the currently applied and the accuracy of novel targeted diagnostic and preventive strategies for EONS.”
Reviewer 2 Report
I really enjoyed reading this very well written article. It is clear and easy to follow.
The manuscript presents a synthetic, critical and didactic overview of the methods that can be used to establish the diagnosis of EONS or the risk for this pathology with the best possible sensitivity and specificity, in order to reduce the antibiotic burden related to EONS. It could even represent a very good book chapter.
The authors discuss the chosen topic from various perspectives: maternal, umbilical and newborn.
I have some recommendations and questions though:
1. I suggest the authors to complete the manuscript with discussions regarding the financial impact of the different strategies discussed, especially those that involve the use of combinations of approaches.
2. did the authors take into account the possibility of using combinations of markers on samples from different sources (mother, umbilical cord, newborn)?
3. Line 51. I think it is Escherichia coli K1.
4. The conclusions must be shortened; the ideas from the introduction must be excluded and the conclusions of the review strictly kept (lines 446-455).
5. The manuscript would be easier to follow if 1-2 figures were added.
Author Response
Reviewer #2
I really enjoyed reading this very well written article. It is clear and easy to follow.
The manuscript presents a synthetic, critical and didactic overview of the methods that can be used to establish the diagnosis of EONS or the risk for this pathology with the best possible sensitivity and specificity, in order to reduce the antibiotic burden related to EONS. It could even represent a very good book chapter. The authors discuss the chosen topic from various perspectives: maternal, umbilical and newborn.
REPLY: First, we thank the reviewer for the very nice words and helpful comments on our manuscript. We are grateful for the provided suggestions and hope to have addressed them with satisfaction.
I have some recommendations and questions though:
- I suggest the authors to complete the manuscript with discussions regarding the financial impact of the different strategies discussed, especially those that involve the use of combinations of approaches.
REPLY: We think this is interesting, however, since a number of approaches are still in the experimental phase, it is difficult to address this. Therefore, we decided not to include this in the manuscript.
- did the authors take into account the possibility of using combinations of markers on samples from different sources (mother, umbilical cord, newborn)?
REPLY: We did and address this issue in the section regarding the calculator: “New iterations of the EOS calculator tool and serial physical examinations will likely lead to some improvement, but it will undoubtedly require use in conjunction with biomarkers and improved diagnostic microbiology tools to reach the goal of treating every newborn with EONS timely, without overtreating those without EONS.” And on GBS: “More extensive studies are needed to determine the diagnostic performance of these genetic virulence markers and to evaluate their potential in combination with current national guidelines for EONS prevention or the neonatal sepsis-calculator”.
Furthermore, in the revised version we have added a sentence in the conclusion section to emphasize this..
NEW: “Finally, future research should also focus on the possibility of using combinations of markers on samples from different sources.”
- Line 51. I think it is Escherichia coliK1.
REPLY: Please see also the review by Simonsen KA, Anderson-Berry AL, Delair SF, Davies HD. Early-onset neonatal sepsis. Clin Microbiol Rev. 2014 Jan;27(1):21-47. doi: 10.1128/CMR.00031-13. PMID: 24396135; PMCID: PMC3910904, page 24
“disease severity is related to the amount and persistence of K1 antigen in the cerebrospinal fluid. Other virulence factors linked to neonatal sepsis include complement resistance mediated by O-lipopolysaccharide and a group of surface proteins which aid in binding and invasion of brain endothelium (including OmpA, IbeA to IbeC, and CNF1)”
Although most likely that K1 antigen plays a major role other virulent factors may play a role. We therefore did not add K1.
- The conclusions must be shortened; the ideas from the introduction must be excluded and the conclusions of the review strictly kept (lines 446-455).
REPLY: According to the reviewers suggestions we shortened the discussion.
- The manuscript would be easier to follow if 1-2 figures were added.
REPLY: In line with the suggestions of the first reviewer we have included 2 figures in the revised manuscript.
Round 2
Reviewer 1 Report
Dear authors,
Thanks for your efforts to improve this manuscript, and for reflecting on the issues that I raised. I would like to commend you on the current text, it has really brushed up in terms of flow, punctuation and conciseness. Furthermore, the two figures are a great addition. Though, there are still some flaws in the text that would benefit from a critical revision.
Major concerns:
- Thanks for affirming previous reviews on this topic. Yet, a good centrality claim is missing in your introduction. More explicitly, which gap did you aim to fill by writing this review? E.g., prevention has not been addressed in similar depth by Fleiss e.a., it would be good to detail this in the introduction so that readers know what your text adds.
- I addressed this earlier: although EONS is in the title, GBS is almost exclusively discussed in the text, in particular about the part on prevention. As you outline, this is the predominant pathogen, but other bacteria are responsible for EONS up to 40% (E.g., Sgro M e.a. PMID: 22547941). Therefore, please change the title to "Group B Streptococcus early-onset disease" instead of "EONS".
- Lines 119-122: please disclose which genotypes are over-represented in invasive disease (whether in the text or in a separate table).
- You raise the concern of inadequate bcx volumes (whether that is too low or high). Please describe what the ideal/recommended volume is, and how much evidence there is for that recommendation.
- A very common discussion point re. GBS is not covered in the text: indications and timing for lumbar punctures in suspected cases. Do the authors have any recommendations on tackling this?
Minor comments:
- Line 27-29: "due to clinical presentation": would be better written as "non-specific symptoms"
- Line 31-33: "In this review ... is discussed": please put the preventive part before dx- in line with the title.
- Lines 201-202: Please clarify that this is postnatal sampling.
- Line 206: Please add references following this enumeration.
- Lines 388-391: Please rephrase this sentence- there are no preventive strategies that could increase the sensitivity of EONS detection.
Round 3
Reviewer 1 Report
Thanks for the new revision. I must say that the quality has improved, but I still detected inaccuracies that must be improved before it can be considered for publication.
1. GBS prevention: "Those who are GBS carriers 86 receive IAP directly and are then tested after treatment [23]".
* First, the abbreviation IAP has not been mentioned before in the text, so it should have been specified.
* But more importantly, re-testing after intrapartum prophylaxis is (at least) highly unusual. What therapeutic implication would that have? It is for sure not recommended by the referenced ACOG guideline.
* An important cause of false negative maternal GBS screening is sampling outside an interval of 5 weeks before delivery. Is there a reason why that is not covered in this text?
2. Host-pathogen interaction: "Examples of virulence factors are α-like 118 proteins that enable cell invasion, the polysaccharide capsule that causes host immune 119 evasion and the hyper-virulent GBS surface-anchored adhesin protein (HvgA) that makes 120 the crossing of the blood–brain barrier possible."
* This statement should be underpinned by a reference.
"Newborns whose mothers have low levels of 135 GBS-protective antibodies or newborns born preterm are at higher risk of invasive GBS 136 disease."
* And this sentence too.
3. Blood and molecular culture techniques
* As far as I know, there is no such a thing as a molecular culture. Therefore, I would rephrase this subtitle.
* An ideal neonatal blood volume of 1.5 ml has been recommended by some authors. Others have argued that increasing the blood volume for bcx could even decrease Se. Please outline on this.
